# Cardiac Implantable Electronic Devices in Hemodialysis and Chronic Kidney Disease Patients—An Experience-Based Narrative Review

**DOI:** 10.3390/jcm10081745

**Published:** 2021-04-17

**Authors:** Krzysztof Nowak, Mariusz Kusztal

**Affiliations:** 1Department of Heart Diseases, Wroclaw Medical University, 50-556 Wroclaw, Poland; 2Department of Nephrology and Transplantation Medicine, Wroclaw Medical University, 50-556 Wroclaw, Poland; mariusz.kusztal@umed.wroc.pl

**Keywords:** cardiovascular implantable electronic device, chronic kidney disease, hemodialysis, vein stenosis

## Abstract

Cardiovascular implantable electronic devices (CIEDs) are a standard therapy utilized for different cardiac conditions. They are implanted in a growing number of patients, including those with chronic kidney disease (CKD) and end-stage kidney disease (ESKD). Cardiovascular diseases, including heart failure and malignant arrhythmia, remain the leading cause of mortality among CKD patients, especially in ESKD. CIED implantation procedures are considered minor surgery, typically with transvenous leads inserted via upper central veins, followed by an impulse generator introduced subcutaneously. A decision regarding optimal hemodialysis (HD) modality and the choice of permanent vascular access (VA) could be particularly challenging in CIED recipients. The potential consequences of arteriovenous access on the CIED side are related to (1) venous hypertension from lead-related central vein stenosis and (2) the risk of systemic infection. Therefore, when creating permanent vascular access, the clinical scenario may be complicated by the CIED presence on one side and the lack of suitable vessels for arteriovenous fistula on the contralateral arm. These factors suggest the need for an individualized approach according to different clinical situations: (1) CIED in a CKD patient; (2) CIED in a patient on hemodialysis CIED; and (3) VA in a patient with CIED. This complex clinical conundrum creates the necessity for close cooperation between cardiologists and nephrologists.

## 1. Introduction

Cardiovascular implantable electronic devices (CIEDs) are a standard therapy utilized for different cardiac conditions. Primarily, these devices are used for the management or prevention of cardiac dysrhythmias and heart failure therapy. They are implanted in a growing number of patients, including those with chronic kidney disease (CKD) and end-stage kidney disease (ESKD). The reported presence of this permanent therapy in hemodialyzed patients exceeds 10%, including an implantable cardioverter-defibrillator (ICD) in 6.1% and a pacemaker (PM) in 4.4% [1]. Similarly, in Atlanta Emory University Hospital data, ESKD was present in 184 (5.3%) of 3453 patients receiving an ICD in 2006–2014 [2]. This reflects the notion that CKD and various cardiac diseases are common comorbidities in an aging population. In CKD patients above 66 years of age, almost two-thirds (64.5%) have concomitant cardiovascular disease (CVD) compared with approximately one-third (32.4%) in a group free of CKD. Specifically, the prevalence of heart failure and myocardial infarction is four times as common among individuals with CKD [3]. Cardiovascular diseases, including heart failure and malignant arrhythmia, remain the leading cause of mortality among CKD patients, especially in ESKD [4]. On the other hand, even early CKD stage increases the risk of morbidity and mortality in various cardiac conditions, particularly heart failure [5]. The presence of CKD strongly influences the clinical management of cardiac conditions, and limits the use of standard therapies [6]. Furthermore, advanced CKD and ESKD are often exclusion criteria in clinical trials of cardiovascular drugs and devices [7].

CIED implantation procedures are considered minor surgery. The most common CIED placement is transvenous lead insertion via puncture of the subclavian/axillary vein or a cephalic vein cut-down method, followed by implantation of an impulse generator in a subcutaneous pocket. The procedure carries some immediate and long-term risks, including pneumothorax, hematoma, infection, and lead-related problems. A decision regarding optimal hemodialysis (HD) modality and the choice of permanent vascular access (VA) can be particularly challenging in CIED recipients. The feasibility of arteriovenous fistula (AVF) creation on the forearm, the gold standard of vascular access (with individual exceptions) for hemodialysis, requires patency of both the artery and vein with adequate size and elasticity. Transvenous CIED lead presence is associated with central vein stenosis. The frequency of severe stenosis or ipsilateral occlusion varies from 11 to 36% [8,9,10]. Therefore, when creating permanent vascular access, a clinical scenario may be complicated by CIED presence on one side and the lack of suitable vessels for arteriovenous fistula on the contralateral arm. This is especially true if hemodialysis is the only available modality of renal replacement therapy. The potential consequences of arteriovenous access on the CIED side are related to (1) venous hypertension from lead-related central vein stenosis and (2) the risk of systemic infection (lead-associated endocarditis). These dangers suggest an individual approach in access creation, including additional interventions, like percutaneous venoplasty, removal and replacement of the CIED system in an alternate site, or creation of alternative arteriovenous access.

This often complex clinical conundrum creates the necessity for close cooperation between cardiologists and nephrologists. Such an interchange should start even before the decision about CIED implantation in a patient with known CKD is made.

## 2. CIED Evidence in CKD Patients

Indications for PM therapy in patients with CKD are similar to the general population. According to current guidelines, PMs should be considered in patients with symptomatic, persistent, or intermittent bradycardia [11].

Improvement of clinical outcomes in patients with CKD treated with an ICD is not apparent. Survival benefit related to ICD therapy in sudden cardiac death (SCD) survivors was observed in dialysis patients [12,13]. This observation encourages ICD implantation for secondary SCD prevention. Despite the fact that the risk of SCD in this patient group is estimated at an annual rate of nearly 6%, there are still doubts about its benefits in the primary prevention of SCD [14]. Such benefits are well-documented in patients with heart failure and reduced left ventricular ejection fraction (LVEF ≤ 35%) [15]. In a metanalysis of three ICD trials, MADIT I, MADIT II, and SCD-HeFT, Pun et al. found that the survival benefit associated with the ICD is diminished in patients with low baseline eGFR [16]. In another publication, there was no difference in mortality following ICD implantation (for primary prevention of SCD) between dialysis patients and matched dialysis recipient controls who did not receive an ICD (HR = 0.94, 95% CI: 0.67–1.31, log-rank *p* = 0.71) [17]. In a retrospective analysis of the MADIT II trial, no benefits related to implanted cardioverter defibrillators were shown among patients with an estimated glomerular filtration rate (eGFR) < 35 mL/min/1.73 m^2^ [18]. According to the recently presented results of the prematurely terminated Implantable Cardioverter-Defibrillator in Dialysis Patients Trial (the sole prospective randomized study investigating the value and safety of transvenous ICD implantation to prevent SCD in dialysis patients with a left ventricular ejection fraction ≥ 35%), prophylactic ICD therapy did not reduce the rate of SCD or all-cause mortality, which remained high [19]. These results could be related to the nature of arrhythmias observed in chronic HD patients. In published data on long-term monitoring of HD patients with implantable loop recorders, the prevalence of bradycardia/asystole events as a potential underlying cause of SCD exceeded the frequency of malignant ventricular arrhythmias [20]. Moreover, patients with ESKD receiving ICDs have a significantly increased risk of developing procedure-related complications [21]. On the other hand, in some studies, a potentially beneficial effect of ICDs in this patient population was described [22,23,24].

Thankfully, cardiac resynchronization therapy (CRT) outcomes in patients with systolic heart failure and CKD are clearly more positive, despite a more complex implantation procedure [24,25,26]. Effective left ventricle reverse remodeling plays a vital role in preserving renal function, regardless of the degree of baseline renal insufficiency [27]. Improvement of renal function emphasizes the importance of cardiorenal interaction, providing another mechanistic argument for CRT’s beneficial effects [28]. Thus, CRT should be offered to CKD patients, if indicated, as expected benefits outweigh the risks associated with this procedure.

In the next part of this narrative review, different clinical issues of CKD/ESKD are discussed. There are no randomized controlled trials (RCTs) on optimal vascular access selection for dialysis in patients with CIEDs. In fact, conducting a properly designed RCT in the ESKD population with coexisting heart failure/arrhythmia and very limited survival is almost impossible. Hence, the proposed ways of proceeding can be viewed as pragmatic approaches based chiefly on the authors’ clinical experience.

### 2.1. CIED Procedure in a CKD Patient

Patients with CKD can be characterized by various risks for the progression to end-stage kidney failure. In general, common clinical variables like older age, male sex, lower eGFR (especially with a decline rate > 5 mL/min/year), albuminuria level, albumin-to-creatinine ratio, systolic blood pressure, smoking status, and presence of diabetes mellitus increase the probability of early occurrence of ESKD [29,30]. Several prediction tools are available. They include, for instance, the four- or eight-variable Kidney Failure Risk Equation [31,32] with an online calculator [33]. Moreover, in every CKD patient considered for CIED implantation, the risk of contrast-induced nephropathy (CIN) due to potential angiography or contrast-enhanced computed tomography should be evaluated [30,31].

In the potential candidates for renal replacement therapy with vascular access creation, the preservation of all peripheral and central veins is of vital importance. Therefore, the decision-making process of CIED implantation requires meticulous planning that includes careful consideration of the following: (1) the type and possible location of the device, taking into account vascular anatomy (previous catheters or infected devices, AVF presence); (2) the patient’s expectations (including prognosis, quality of life); and (3) patients with CKD grade 4 or 5 should be referred to a vascular access specialist (surgeon, nephrologist, or radiologist with experience in vascular access) for the examination of vascular anatomy, in order to decide which site will be optimal for future AV fistula/graft placement. A physical examination, obtaining the history of previous vessel cannulation, and ultrasound mapping (including jugular vein size and configuration) should be sufficient in the majority of candidates for CIED and HD. Venography, including central veins, should also be performed in case of any doubts or a history of previous catheters. Consulting a vascular access specialist should provide the cardiologist with vascular recommendations for the theoretic feasibility of AVF/AVG formation, which should be considered when planning the CIED surgery. Standard placement of CIED into the left pectoral region in case of quickly progressing CKD patients should always be discussed, and alternative options considered (Table 1) to avoid vascular conflict and further complications.

Patients with chronic renal disease have an increased risk of device infection due to immune alterations related to the disease and possible bacteriemia from frequent vascular access and dialysis catheters [34]. This complication usually requires complete CIED extraction. Chronic kidney disease is associated with accelerated vascular calcification, i.e., the pathological deposition of minerals in the form of calcium phosphate salts in the vascular tissues. It is an active and complex process related to abnormal calcium and phosphorus levels and subsequent osteogenic differentiation of vascular smooth muscle cells closely resembling bone formation [34]. Widespread vascular calcification, together with a higher probability of lead–vessel or lead–lead binding found in CKD patients, particularly in those with end-stage renal failure requiring dialysis, may make transvenous lead extraction procedures challenging, and is associated with a higher risk of complications [35].

Attention should be paid to patients on peritoneal dialysis (PD), especially those with longer predicted survival (>three years), because, in approximately 20–30%, there may be a need for permanent HD (loss of effectiveness of PD with time; multiple peritonitis episodes, etc.) or temporal HD (peritonitis, hernia surgery). If PD was commenced as a chosen way of renal replacement therapy, one must remember that, in some cases, PD might be not effective, and cannot be continued. Therefore, if the patient’s condition allows (the indication for CIED is nonurgent and implantation can wait four weeks), a more appropriate order of action would be PD catheter implantation, followed by a peritoneal equilibration test (PET) after four weeks to assess peritoneal transport characteristics. An uncomplicated/efficient PD course allows for CIED implantation. Later on, the need to switch the dialysis modality is still substantial. In a recent case–control study, low dialysis treatment adequacy, low albumin level, a higher number of hospitalizations, and history of peritonitis were factors associated with PD transfer to HD [36]. With this approach, priority is given to prudent access selection.

Proposed treatment options in CKD patients with a high risk of ESKD or on dialysis are summarized in Figure 1.

### 2.2. CIEDs in a Patient on Hemodialysis

It is generally recommended to place the CIED on the contralateral upper limb relative to the arteriovenous VA or central vein catheter (CVC) [37].

There are some options to avoid transvenous CIED implantation in HD patients. These include: (1) a CIED with leads implanted epicardially; (2) subcutaneous ICD (S-ICD) for SCD prevention; (3) leadless PM for bradycardia therapy; or (4) left ventricle endocardial pacing with the leadless ultrasound-based technology for CRT (Table 1).

The implantation of CIED employing the epicardial leads eliminates the collision trajectory with VA. Epicardial leads traverse through the subcutaneous tissue, and do not require vascular puncture. In this context, vascular injury or pneumothorax are rare complications of the procedure [38]. Other advantages include reduced radiation exposure and not using intravenous contrast. However, their placement is much more invasive than transvenous implantable devices, and requires both general anesthesia and a cardiothoracic surgeon, especially when a conventional procedure with sterno- or thoracotomy is considered [39]. Recently emerging minimally invasive surgical implantation techniques, like minithoracotomy, video thoracoscopy, and robotic surgery, reduce the risk of sterno- and thoracotomy, and allow all pacing modes, including resynchronization therapy [40,41,42,43]. Efficacy and mortality were usually comparable to patients with endocardial implanted systems, especially if steroid-eluting epicardial leads were used [44,45]. Moreover, new percutaneously implantable epicardial systems have been successfully tested in animal models [46,47]. Some concerns are raised in terms of long-term outcomes. In some published data concerning mainly children or adults with congenital heart disease, the longevity of epicardial pacing systems was inferior to endocardial systems [48,49,50].

As an alternative to the pectoral and epicardial approaches for pacing, the femoral route has also been successful. Technically, leads are introduced into the femoral vein (usually, the great saphenous venipuncture is preferred over the cut-down to avoid femoral vein thrombosis), and the incision is made in the lower abdomen to position the generator [51]. A better approach involves puncture of the iliac vein. The safest technique begins with the puncture of the femoral vein (with or without ultrasound guidance) and passing along guidewire up to the inferior vena cava (IVC). The guidewire is used as a target for the puncture of the ipsilateral iliac vein (above the inguinal ligament). Once the iliac vein is accessed, a second guidewire is inserted and advanced to the IVC or right atrium. Additional iliac vein access may be gained the same way for dual-chamber, or less often, for cardiac resynchronization therapy [52,53,54]. Besides technical issues, this solution still uses a transvenous lead.

Leadless pacing is an attractive alternative to transvenous pacemaker implantation. Reported multicenter experience with leadless pacing using a Medtronic Micra™ device (Medtronic, Mounds View, Minnesota, USA) showed that, among 2819 implantations, about 7% were performed in HD patients [55]. This group was characterized by acceptable procedural safety, low long-term pacing thresholds with good sensing, and battery life. Moreover, no patients developed a device-related infection. The Micra device is inserted via the femoral vein and advanced to the right ventricular septum, where fixation is achieved via nitinol tines. The leadless device offers a rate-responsive ventricular pacing mode. The Micra™ AV leadless pacing system (Medtronic, Mounds View, MN, USA) can provide efficient atrioventricular synchrony via a unique pacing algorithm that relies on identifying mechanical atrial contraction [56].

Drawbacks of the transvenous ICDs were presented above. Recently, the subcutaneous ICD (S-ICD; Emblem™; Boston Scientific, Marlborough, MA, USA) has entered into clinical practice as an important alternative, especially among patients with limited vascular access, increased risk of infection, and a structurally normal heart with no need for pacing [57]. Recent reports indicated a considerable increase in the utilization of an S-ICD in CKD patients on long-term hemodialysis [58]. Insertion of an S-ICD is a minimally invasive procedure. By virtue of leaving the venous system untouched, this approach might offer the advantage of a reduced risk of central venous stenosis and infection over an endocardial ICD with transvenous leads [59]. Importantly, this device requires preimplant screening to ensure appropriate sensing and reduce the risk of inappropriate shocks [60]. Besides challenging vascular access (central veins stenosis, occlusion, AVF, or CVC for HD) and complications avoidance, the following indications for S-ICD placement should be considered: prior complications with transvenous ICDs, infection (bacteremia), and the young age of the ICD patient. Results published for randomized clinical trials support S-ICD use [61,62]. Koman et al. reported data from the follow-up of 18 HD patients and 78 non-HD patients with implanted S-ICD. HD patients implanted with S-ICD had similar procedural outcomes and inappropriate shock frequency. All appropriate shocks were successful in terminating ventricular tachyarrhythmias in both groups. There was no device or blood stream–related infection in HD patients, compared to nearly 7% in the non-HD group [63]. This seemingly paradoxical difference did not reach statistical significance. Similar results were presented by El-Chami et al. [64]. Unfortunately, in February 2021, the US Food and Drug Administration issued two class 1 recalls of Emblem™ (Boston Scientific, Natick, MA, USA). The first one involved an increased risk of lead fracture that could prevent the ICD from delivering life-saving therapy. The second one concerned a fault that allows moisture to enter the generator, causing it to shortcircuit when it tries to shock patients [65,66].

CRT remains challenging in ESKD patients. CRT systems usually consist of three implanted transvenous leads: two positioned in the right heart and the last one on the left ventricle (LV) epicardium via coronary sinus branches. There is a unique option with a wireless LV endocardial pacing system (WiSE-CRT™, EBR Systems, Sunnyvale, CA, USA). The WiSE system synchronizes LV pulses to the RV pulse by sensing a specified pulse width from a conventional or leadless pacing device. It then delivers focused ultrasound energy from an ultrasound pulse generator implanted subcutaneously (in one of the fourth to sixth intercostal spaces lateral to the left parasternal border) to an LV electrode (within 2–5 milliseconds). The LV electrode delivers electrical stimulation to the LV endocardial surface of the heart by transducing acoustic energy. The LV receiver electrode is fully endothelialized after four weeks, avoiding the need for long-term anticoagulation [67]. In a multicenter registry of 90 patients implanted with the WiSE-CRT system, successful implantation and CRT delivery was achieved in 94.4% of patients, while 70% of patients experienced an improvement in heart failure symptoms [68]. Montemerlo et al. presented a case of fully leadless resynchronization therapy delivered trans-septally with a combination of Micra™ and WiSE-CRT™ systems [69]. Carabelli et al. published their experiences in a case series with totally leadless CRT successfully delivered with a similar combination [70]. All eight WiSE-CRT devices could successfully detect the Micra pacing output and be trained to deliver synchronous LV endocardial pacing. Acute QRS reduction following WiSE-CRT implantation was observed in all eight patients. Six months after WiSE-CRT implantation, there was a significant increase in the LV ejection fraction (28.43 ± 8.01% vs. 39.71 ± 1.89%; *p* = 0.018) [70].

### 2.3. Vascular Access in a Patient with a CIED

As mentioned above, in non-dialysis patients, the insertion of transvenous lead(s) might result in central vein stenosis [8,9]. Central vein stenosis (CVS) is asymptomatic in most patients, but this is not the case in the HD population. Here, CVS incidence appears to be higher because of the abnormal hemodynamics associated with increased blood flow through the central veins in patients equipped with an AVF. Teruya et al. found central vein stenosis in 10/14 (71%) dialyzed patients with an AVF (no history of CVC insertion) positioned ipsilaterally to CIED [71]. CVS symptoms appeared on average 12.6 months after AVF formation. Thus, AVF/AVG should be created on the upper limb contralaterally to the CIED, and novel devices (like leadless pacemakers, if possible) should be considered in dialysis patients.

However, this is not always feasible due to the limited availability of VA sites and the frequent (and sometimes inevitable) necessity for VA revision due to suboptimal patency. Unexpectedly, according to recently published retrospective studies, both ipsilateral and contralateral VA and CIED placement were related to similar long-term VA patency [72,73]. Nonetheless, because of an increased incidence of systemic complications (e.g., central vein stenosis) [74], we prefer to avoid ipsilateral CIED and VA whenever possible.

The worst possible combination is an ipsilateral CIED together with a central venous catheter (CVC) used for hemodialysis. In such a case, the risk of symptomatic central venous stenosis and the development of endovascular infection with potentially lethal sequelae is multiplied [75,76,77]. This dangerous situation may be temporary (the catheter serving as the access while AVF is maturing) or permanent if no suitable vessel for AVF/AVG is found. Common circumstances leading to CVC use include failure of existing AV access pending revision or construction of new access, failed or dysfunctional peritoneal dialysis, failed kidney transplant, or acute kidney injury. In general, CVC in HD patients with a transvenous CIED should be avoided, and, if necessary, the duration of CVC minimalized. Moreover, the CVC should be placed contralaterally to the transvenous CIED, and tunneling should be performed carefully to avoid entry into the CIED pocket. Awareness of the patient’s pacing dependency (intrinsic heart rate < 30/bpm) during CVC insertion next to the CIED lead is of paramount importance. Thus, CIED function must be assessed before, and ECG monitoring should be available during the procedure. Immediate access to temporary on-site pacing is necessary if maneuvers during CVC placement cause lead displacement and a lack of effective stimulation. The assistance of an interventional radiologist who can perform percutaneous angioplasty (PTA) and help with proper tip location/reposition may be needed for safe catheter reimplantation or exchange while CIED leads are still in situ.

A reassesssment of CIED therapy indications should be performed in patients with limited vascular access. If criteria for CIEDs are no longer met, which is not rare [78], or another approach is possible, lead extraction can be combined with another implantation technique to reestablish and maintain central vein patency.

Proposed decision-making process on vascular access in an ESKD patient with CIED is displayed in Figure 2.

## 3. Conclusions

The co-existence of cardiac implantable devices and arteriovenous fistulae/grafts/tunneled catheters in a patient requiring permanent hemodialysis may create a complex vascular dilemma. Both the vascular access expert and implanting cardiologist must consider possible complications related to their decisions regarding treatment modalities. The ability to predict the risk of the progression to ESKD among potential CIED recipients is of vital importance in the context of the optimal site of implantation/anastomosis, best timing (urgency) of the procedures, appropriate patient selection (predicted survival), and the choice of CIED type (transvenous vs. subcutaneous vs. leadless). Consequently, the idea of establishing a “cardio-renal” team for effective cooperation in this area appears very promising.

## Figures and Tables

**Figure 1 jcm-10-01745-f001:**
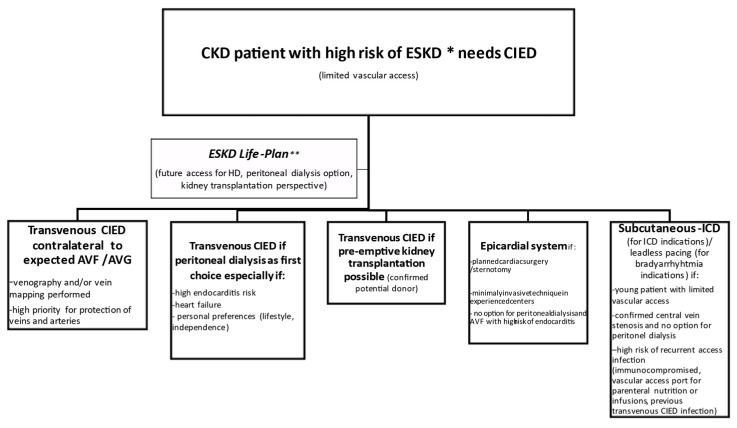
Decision-making on CIED in CKD patients with a high risk of ESKD or on dialysis. CKD—chronic kidney disease, ESKD—end-stage kidney disease, CIED—cardiovascular implantable electronic device, ICD—implantable cardioverter-defibrillator AVF—arteriovenous fistula, AVG—arteriovenous graft. * Estimated risk of progression to ESKD in CKD patients using (four or eight variables) KFRE is available at https://qxmd.com/calculate (accessed on 13 April 2021); ** ESKD Life-Plan approach in concordance with the KDOQI Clinical Practice Guideline for Vascular Access: 2019 Update (https://www.ajkd.org/issue/S0272-6386(20)X0004-7) (accessed on 13 April 2021).

**Figure 2 jcm-10-01745-f002:**
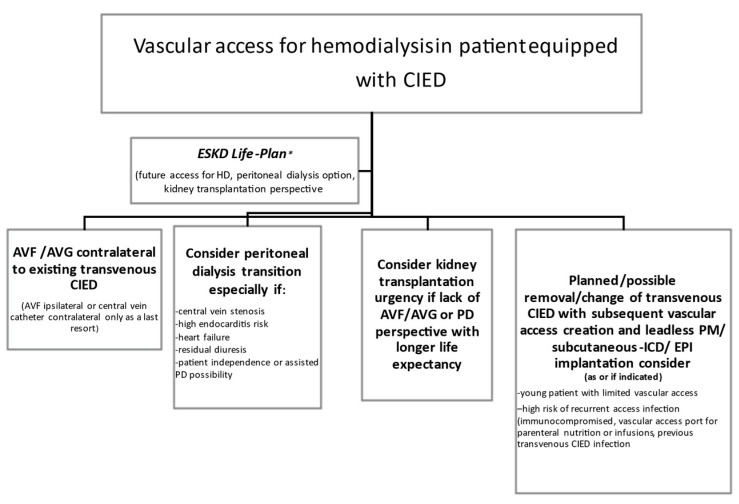
Decision-making on vascular access in an ESKD patient with CIED. AVF—arteriovenous fistula, AVG—arteriovenous graft, ESKD—end-stage kidney disease, CIED—cardiovascular implantable electronic device, ICD—implantable cardioverter-defibrillator, PD—peritoneal dialysis, EPI—epicardial, PM—pacemaker. * ESKD Life-Plan approach in concordance with the KDOQI Clinical Practice Guideline for Vascular Access: 2019 Update (https://www.ajkd.org/issue/S0272-6386(20)X0004-7) (accessed on 13 April 2021).

**Table 1 jcm-10-01745-t001:** Alternative (to standard prepectoral) CIED placement options to consider in patients with CKD/ESKD.

CIED Option	Comment
Leadless pacemaker (endovascular implantation), subcutaneous ICD (S-ICD), endocardial LV pacing based on ultrasound technology	Preferred if indications are limited to pacing or high-energy therapy only; no data regarding long-term outcomes
Surgically implanted epicardial system	When thoracotomy/heart surgery for other reasons is considered; higher risk of lead malfunction; minimally invasive technique available
Low lateral thoracic or axillary implantation	In patients with limited prepectoral options (previous pocket infection); involves axillary vein puncture
Femoral/iliac pacing system	Usually in bilateral subclavian vein occlusion/superior vena cava syndrome; unsuitable in case of poor hygiene (diapers) or local skin infection; risk of lead fracture in walking patients—use of the iliac vein should limit the risk

## Data Availability

Not applicable.

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
