# Peer review of "Cardiac Implantable Electronic Devices in Hemodialysis and Chronic Kidney Disease Patients—An Experience-Based Narrative Review"

_jcm, 2021, doi:10.3390/jcm10081745_

Round 1

Reviewer 1 Report

The topic of implantable medical devices in patients with pre-existing implants is an important one.  The review is of high quality and covers the major concerns of providing implants for the Chronic Kidney Patients (CKD) that need vascular access but who may already have devices that occupy or complicate vascular access in these areas.  The call for clinical trials for vascular access for patients with implants should be considered to optimise patient outcomes.  The review is comprehensive and I make only minor suggestions to improve readability and increase access for non clinicians with an interest in this area.

Minor

1) The role of transverse CEID leads associated with venous stenosis and lack of suitable vessels on the contralateral side is the source of the problem.  A diagram explaining these scenario as the basis of the problem should be elaborated upon and this should be provided earlier  in the introduction of the script.

2) It would be useful to highlight the decision making process for graft over AV fistula or central line access.  How does this AV decision making process change in response to point 1 above ?  A decision making tree or examples of different patient case scenario’s would be useful to improve understanding and the thought processes involved.

Author Response

Dear Editors and Reviewer

Journal of Clinical Medicine

We appreciate for all the suggestions and  comments regarding our manuscript.

After reviewers suggestions we  decided to send our manuscript for professional English editing service. The certificate is included.

Below are all responses that we can offer to the reviewer:

1) The role of transverse CIED leads associated with venous stenosis and lack of suitable vessels on the contralateral side is the source of the problem.  A diagram explaining these scenario as the basis of the problem should be elaborated upon and this should be provided earlier  in the introduction of the script.

2) It would be useful to highlight the decision making process for graft over AV fistula or central line access.  How does this AV decision making process change in response to point 1 above ?  A decision making tree or examples of different patient case scenario’s would be useful to improve understanding and the thought processes involved.

Response:

We added two diagrams with proposed decision algorithm in two scenarios. We aimed to underline early collaboration between implanting cardiologist and nephrologist and  vascular access specialist to draw more attention for patient perspectives. New figures added:

 Figure 1. Decision making on CIED in CKD patient with high risk of ESKD or on dialysis. (line177)  Figure 2. Decision making on vascular access in ESKD patient with CIED. The last diagram take into account clinical or subclinical venous stenosis. (line: 323)

Additional or changed paragraphs have been bolded in the text, references also.

I really appreciate all the suggestions made by editor and reviewer.

Best regards

Krzysztof Nowak, MD, PhD

Author Response

Dear Editors and Reviewer

Journal of Clinical Medicine

We appreciate for all the suggestions and  comments regarding our manuscript.

Following reviewer remark we send the manuscript for professional English editing sevice. The certificate is included. All listed by reviewer sentences has been changed. Thank you for this hard  work.

We appreciate for all “major comments”  with lines indication. All of them has been accepted and changed following reviewer suggestion.

Regarding : The possibility of a systemic infection and device seeding may occur whether permanent vascular access is on the CIED’s ipsilateral or contralateral side.

The systemic infection resulting from bacteriemia, especially in endocarditis history, is serious risk independently from side of central catheter implantation. Catheter end tip and transvenous CIED leads are  in proximity this is why catheter in general should be avoided.

We added appropriate changes in tables and references following your suggestions.

Additional or changed paragraphs have been bolded in the text, references also.

Reviewer 3 Report

The spectrum of alternatives to transvenous CIED implantations is wider than described. There are good epicardial opportunities without full sternotomy; lead longevity is at minimum comparable to transvenous leads.

The largest problem of CIED therapy in patients with chronic renal disease is the much higher risk of severe calcifications around the leads after several months to years which do have an impact on the complexity of lead extraction. 

Author Response

Dear Editors and Reviewer

Journal of Clinical Medicine

We appreciate for all the suggestions and  comments regarding our manuscript.

After reviewers suggestions we  decided to send our manuscript for professional English editing service. The certificate is included.

The spectrum of alternatives to transvenous CIED implantations is wider than described. There are good epicardial opportunities without full sternotomy; lead longevity is at minimum comparable to transvenous leads.

The largest problem of CIED therapy in patients with chronic renal disease is the much higher risk of severe calcifications around the leads after several months to years which do have an impact on the complexity of lead extraction.

Thank you for this remark. We obviously share the opinion that non sternotomy approach is great option but it available in limited centers. We mention this option in recent manuscript.(lines 194-210).  

The extensive calcification problem around leads is another important remark from experienced reviewer. Thank you for that. This point has been developed in revised version with appropriate citation. (line 150-161; ref 34,35).

Additional or changed paragraphs have been bolded in the text, references also.

I really appreciate all the suggestions made by editor and reviewer

Best regards

Krzysztof Nowak, MD, PhD

Round 2

Reviewer 2 Report

The manuscript is improved. The grammar and syntax still require refinement. Attention to the following should improve the manuscript further.

Minor comments:

  1. On lines 13 and 33, change: “applied to” “implanted in”.
  2. On lines 37-39, change: “Similarly to Atlanta Emory University Hospital data, where among 3,453 patients receiving an ICD in 2006–2014, ESKD was present in 184 (5.3%) [2]” to “Similarly at Atlanta’s Emory University Hospital, ESKD was present in 184 (5.3%) of 3,453 patients receiving an ICD during 2006–2014[2]”.
  3. On line 39, change “It” to “This”.
  4. On line 46, change: “the early CKD stage” to “early stage CKD”.
  5. On line 53, delete the word “method”.
  6. On line 57, change: “could” to “can”.
  7. On line 63, delete: “the” twice.
  8. On line 78, change: “ICD” to “an ICD”.
  9. On Line 80, delete: “the”
  10. On line 91, change: “defibrillator” to “defibrillators”.
  11. On lines 116-117, change: “with very limited survival” to “and very limited survival”.
  12. On line 123, change: “presence of diabetes mellitus” to “and presence of diabetes mellitus”.
  13. On lines 125-126, change: “They include the four- or eight-variable Kidney Failure Risk Equation [31, 32] with an online calculator [33], for instance” to “They include, for instance, the four- or eight-variable Kidney Failure Risk Equation [31, 32] with an online calculator [33]”.
  14. On line 132, delete: “aspects”.
  15. In table 1, change: “vena cava superior syndrome” to “superior vena cava syndrome”.
  16. On line 160, change: “related to” to “associated with”.
  17. On lines 165-168, change:  “If PD was commenced as a chosen way of renal replacement therapy and the indication for CIED is nonurgent (can wait four weeks), one must remember that PD in some cases might be not effective and thus cannot be continued” to  “If PD was commenced as a chosen way of renal replacement therapy, one must remember that, in some cases, PD might be not effective and cannot be continued”.
  18. On lines 168-170, change: “If the patient’s condition allows, a more appropriate order of action would be PD catheter implantation followed by a peritoneal equilibration test (PET) after four weeks to assess peritoneal transport characteristics” to “Therefore, if the patient’s condition allows (the indication for CIED is non-urgent and implantation can wait four weeks), a more appropriate order of action would be PD catheter implantation followed by a peritoneal equilibration test (PET) after four weeks to assess peritoneal transport characteristics”.
  19. On line 170, change: “Uncomplicated/efficient” to “An uncomplicated/efficient”.
  20. On line 188, change: “CIED” to “the CIED”.
  21. On line 194, change: “CIED” to “a CIED”.
  22. On line 197, delete: “risk”.
  23. On line 199, delete: “additionally”.
  24. On line 201, change: “The recently emerged” to “Recently emerging”.
  25. On line 202, delete: “thorough”. It is not necessary to replace it with the word “through”.
  26. On line 203, delete: “overt”.
  27. On line 210, change: “endocardial” to “endocardial systems”.
  28. On line 213, you write “cut-off”. Do you mean “cut-down”?
  29. On lines 215-221, change: “A better approach involves puncture of the iliac vein (above the inguinal ligament) or the puncture of the femoral vein (with or without ultrasound guidance) and passing along guidewire up to the inferior vena cava (IVC). The guidewire is used as a targetfor puncture of the ipsilateral iliac vein. Once the iliac vein is accessed, a second guidewire is inserted and advanced to the IVC or right atrium. Additional iliac vein access may be gained the same way for dual-chamber, or less often, for cardiac resynchronization therapy [52–54]” to “A better approach involves puncture of the iliac vein. The safest technique begins with puncture of the femoral vein (with or without ultrasound guidance) and passing a long guidewire up to the inferior vena cava (IVC). The guidewire is used as a target for puncture of the ipsilateral iliac vein (above the inguinal ligament). Once the iliac vein is accessed, a second guidewire is inserted and advanced to the IVC or right atrium. Additional iliac vein access may be gained the same way for dual-chamber, or less often, for cardiac resynchronization therapy [52–54]”.
  30. On line 229, change: “the ventricle rate-responsive mode” to “a rate-responsive ventricular pacing mode”.
  31. On line 270, change: “with implanted” to “implanted with”.
  32. On lines 306-307, change “transvenous” to  “a transvenous”.
  33. On line 318, change: “CIED therapy indications reassessment” to “Reassessment of CIED therapy indications”.
  34. On lines 333-334, change: “provokes a conflicting vascular situation” to may create a complex vascular dilemma”.

Major comments:

  1. If figure 1 is reproduced or adapted from the sources listed, permission to do so must be obtained and acknowledged in the figure legend.
  2. On line 202: Subxiphoid incisions have been used for more than 20 years and should not be considered emerging technology.
  3. Reference 46 is a canine study and reference 47 is a porcine and canine study. Therefore, on line 207, “are developed” should be changed to “have been successfully tested in animal models”.
  4. The notion that CVS is related to increased blood flow through the central veins seems counter-intuitive. Is this a “wear and tear” phenomenon? 
  5. Figure 2 is difficult to see. White rectangles with black print would be a better choice.

Author Response

Minor comments:

  1. On lines 13 and 33, change: “applied to” “implanted in”.
  2. On lines 37-39, change: “Similarly to Atlanta Emory University Hospital data, where among 3,453 patients receiving an ICD in 2006–2014, ESKD was present in 184 (5.3%) [2]” to “Similarly at Atlanta’s Emory University Hospital, ESKD was present in 184 (5.3%) of 3,453 patients receiving an ICD during 2006–2014[2]”.
  3. On line 39, change “It” to “This”.
  4. On line 46, change: “the early CKD stage” to “early stage CKD”.
  5. On line 53, delete the word “method”.
  6. On line 57, change: “could” to “can”.
  7. On line 63, delete: “the” twice.
  8. On line 78, change: “ICD” to “an ICD”.
  9. On Line 80, delete: “the”
  10. On line 91, change: “defibrillator” to “defibrillators”.
  11. On lines 116-117, change: “with very limited survival” to “and very limited survival”.
  12. On line 123, change: “presence of diabetes mellitus” to “and presence of diabetes mellitus”.
  13. On lines 125-126, change: “They include the four- or eight-variable Kidney Failure Risk Equation [31, 32] with an online calculator [33], for instance” to “They include, for instance, the four- or eight-variable Kidney Failure Risk Equation [31, 32] with an online calculator [33]”.
  14. On line 132, delete: “aspects”.
  15. In table 1, change: “vena cava superior syndrome” to “superior vena cava syndrome”.
  16. On line 160, change: “related to” to “associated with”.
  17. On lines 165-168, change:“If PD was commenced as a chosen way of renal replacement therapy and the indication for CIED is nonurgent (can wait four weeks), one must remember that PD in some cases might be not effective and thus cannot be continued” to  “If PD was commenced as a chosen way of renal replacement therapy, one must remember that, in some cases, PD might be not effective and cannot be continued”.
  18. On lines 168-170, change: “If the patient’s condition allows, a more appropriate order of action would be PD catheter implantation followed by a peritoneal equilibration test (PET) after four weeks to assess peritoneal transport characteristics” to “Therefore, if the patient’s condition allows (the indication for CIED is non-urgent and implantation can wait four weeks), a more appropriate order of action would be PD catheter implantation followed by a peritoneal equilibration test (PET) after four weeks to assess peritoneal transport characteristics”.
  19. On line 170, change: “Uncomplicated/efficient” to “An uncomplicated/efficient”.
  20. On line 188, change: “CIED” to “the CIED”.
  21. On line 194, change: “CIED” to “a CIED”.
  22. On line 197, delete: “risk”.
  23. On line 199, delete: “additionally”.
  24. On line 201, change: “The recently emerged” to “Recently emerging”.
  25. On line 202, delete: “thorough”. It is not necessary to replace it with the word “through”.
  26. On line 203, delete: “overt”.
  27. On line 210, change: “endocardial” to “endocardial systems”.
  28. On line 213, you write “cut-off”. Do you mean “cut-down”?
  29. On lines 215-221, change: “A better approach involves puncture of the iliac vein (above the inguinal ligament) or the puncture of the femoral vein (with or without ultrasound guidance) and passing along guidewire up to the inferior vena cava (IVC). The guidewire is used as a targetfor puncture of the ipsilateral iliac vein. Once the iliac vein is accessed, a second guidewire is inserted and advanced to the IVC or right atrium. Additional iliac vein access may be gained the same way for dual-chamber, or less often, for cardiac resynchronization therapy [52–54]” to “A better approach involves puncture of the iliac vein. The safest technique begins with puncture of the femoral vein (with or without ultrasound guidance) and passing a long guidewire up to the inferior vena cava (IVC). The guidewire is used as a target for puncture of the ipsilateral iliac vein (above the inguinal ligament). Once the iliac vein is accessed, a second guidewire is inserted and advanced to the IVC or right atrium. Additional iliac vein access may be gained the same way for dual-chamber, or less often, for cardiac resynchronization therapy [52–54]”.
  30. On line 229, change: “the ventricle rate-responsive mode” to “a rate-responsive ventricular pacing mode”.
  31. On line 270, change: “with implanted” to “implanted with”.
  32. On lines 306-307, change “transvenous” to“a transvenous”.
  33. On line 318, change: “CIED therapy indications reassessment” to “Reassessment of CIED therapy indications”.
  34. On lines 333-334, change: “provokes a conflicting vascular situation” to may create a complex vascular dilemma”.

The authors change text according to these comments.

Major comments:

  1. If figure 1 is reproduced or adapted from the sources listed, permission to do so must be obtained and acknowledged in the figure legend.

Figure 1is our proposal elaborated for  our patients admitted to our hospital. There is lack of official society (cardiac/renal/vascular) recommendation or algorithm on this problem.

  1. On line 202: Subxiphoid incisions have been used for more than 20 years and should not be considered emerging technology.

The authors decide  to change text in accordance with the remark

  1. Reference 46 is a canine study and reference 47 is a porcine and canine study. Therefore, on line 207, “are developed” should be changed to “have been successfully tested in animal models”.

Of course, these papers present animal study.

  1. The notion that CVS is related to increasedblood flow through the central veins seems counter-intuitive. Is this a “wear and tear” phenomenon? 

Line 290-292; 

There are subclinical (non dialysis patients without AVF/catheter) SVC in association with CIED and clinical/symptomatic in those with AVF and CIED due to significant obstruction in venous outflow route (of course AVF cause increase venous blood flow , usually >500ml/min).

However we  accept  that this is still hypothetic (unless large detail comparative study conduction ).

Similar hypotheses are mentioned by others:

”patient characteristics(…) and elevated venous pressure during HD concurred to vascular damage and constituted a prerequisite for the steno-thrombosis in the presence of catheter-induced trauma”

“ A possible explanation for SVC development beside vein injury during implantation is that repeated traumatization by leads in the vein wall is responsible for the progressive fibrotic stenosis just above the right atrium. Endothelialization of pacing leads can further reduce the luminal diameter of the vessel, leading to a clinically significant CVS”

Pacilio M, Borrelli S, Conte G, Minutolo R, Musumeci A, Brunori G, Veniero P, De Falco V, Provenzano M, De Nicola L, Garofalo C: Central Venous Stenosis after Hemodialysis: Case Reports and Relationships to Catheters and Cardiac Implantable Devices. Cardiorenal Med 2019;9:135-144. doi: 10.1159/000496065

  1. Figure 2 is difficult to see. White rectangles with black print would be a better choice.

The authors change Figures according Reviewer advice.